# Concept Embedding Models:
# Beyond the Accuracy-Explainability Trade-Off

**Mateo Espinosa Zarlenga**[*]
University of Cambridge
me466@cam.ac.uk

**Pietro Barbiero**[*]
University of Cambridge
pb737@cam.ac.uk

**Gabriele Ciravegna**
Université Côte d'Azur, Inria, CNRS,
I3S, Maasai, Nice, France
gabriele.ciravegna@inria.fr

**Giuseppe Marra**
KU Leuven
giuseppe.marra@kuleuven.be

**Francesco Giannini**
University of Siena
francesco.giannini@unisi.it

**Michelangelo Diligenti**
University of Siena
diligmic@diism.unisi.it

**Zohreh Shams**
Babylon Health
University of Cambridge
zs315@cam.ac.uk

**Frederic Precioso**
Université Côte d'Azur, Inria, CNRS,
I3S, Maasai, Nice, France
fprecioso@unice.fr

**Stefano Melacci**
University of Siena
mela@diism.unisi.it

**Adrian Weller**
University of Cambridge
Alan Turing Institute
aw665@cam.ac.uk

**Pietro Lio**
University of Cambridge
pl219@cam.ac.uk

**Mateja Jamnik**
University of Cambridge
mateja.jamnik@cl.cam.ac.uk

## Abstract

Deploying AI-powered systems requires trustworthy models supporting effective human interactions, going beyond raw prediction accuracy. Concept bottleneck models promote trustworthiness by conditioning classification tasks on an intermediate level of human-like concepts. This enables human interventions which can correct mispredicted concepts to improve the model's performance. However, existing concept bottleneck models are unable to find optimal compromises between high task accuracy, robust concept-based explanations, and effective interventions on concepts—particularly in real-world conditions where complete and accurate concept supervisions are scarce. To address this, we propose Concept Embedding Models, a novel family of concept bottleneck models which goes beyond the current accuracy-vs-interpretability trade-off by learning interpretable high-dimensional concept representations. Our experiments demonstrate that Concept Embedding Models (1) attain better or competitive task accuracy w.r.t. standard neural models without concepts, (2) provide concept representations capturing meaningful semantics including and beyond their ground truth labels, (3) support test-time concept interventions whose effect in test accuracy surpasses that in standard concept bottleneck models, and (4) scale to real-world conditions where complete concept supervisions are scarce.

---

[*]Equal contribution

36th Conference on Neural Information Processing Systems (NeurIPS 2022).

# 1 Introduction

What is an *apple*? While any child can explain what an "apple" is by enumerating its characteristics, deep neural networks (DNNs) fail to explain what they learn in human-understandable terms despite their high prediction accuracy [1]. This accuracy-vs-interpretability trade-off has become a major concern as high-performing DNNs become commonplace in practice [2–4], thus questioning the ethical [5, 6] and legal [7, 8] ramifications of their deployment.

Concept bottleneck models (CBMs, [9], Figure 1a) aim at replacing "black-box" DNNs by first learning to predict a set of concepts, that is, "interpretable" high-level units of information (e.g., "color" or "shape") [10] provided at training time [11, 12], and then using these concepts to learn a downstream classification task. Predicting tasks as a function of concepts engenders user trust [13] by allowing predictions to be explained in terms of concepts and by supporting human interventions, where at test-time an expert can correct a mispredicted concept, possibly changing the CBM's output. That said, concept bottlenecks may impair task accuracy [9, 14], especially when concept labels do not contain all the necessary information for accurately predicting a downstream task (i.e., they form an "incomplete" representation of the task [15]), as seen in Figure 1b. In principle, extending a CBM's bottleneck with a set of unsupervised neurons may improve task accuracy, as observed by Mahinpei et al. [14]. However, as we will demonstrate in this work, such a hybrid approach not only significantly hinders the performance of concept interventions, but it also affects the interpretability of the learnt bottleneck, thus undermining user trust [13]. Therefore, we argue that novel concept-based architectures are required to overcome the current accuracy/interpretability pitfalls of CBMs, thus enabling their deployment in real-world settings where concept annotations are likely to be incomplete.

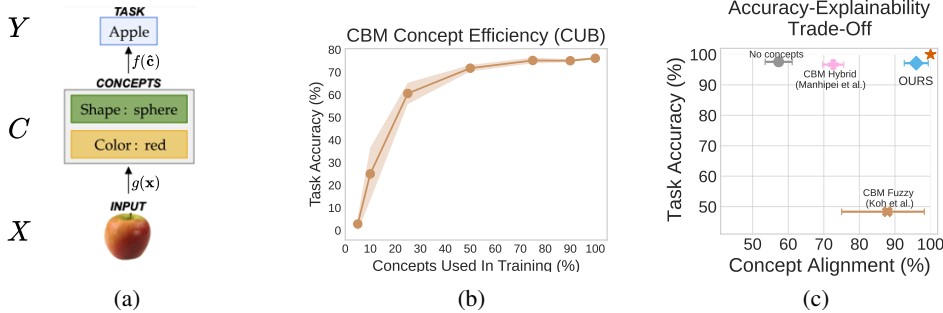

Figure 1: (a) A concept bottleneck model, (b) task accuracy after using only a fraction of total concept annotations to train a CBM on CUB [16] and, (c) the accuracy-vs-interpretability trade-off (corner star represents the optimal trade-off).

In this paper, we propose Concept Embedding Models (CEMs), a novel concept bottleneck model (described in Section 3) which overcomes the current accuracy-vs-interpretability trade-off found in concept-incomplete settings (as shown in Figure 1c). Furthermore, we introduce two new metrics for evaluating concept representations (Section 4) and use them to help understand why our approach circumvents the limits found in the current state-of-the-art CBMs (Section 5). Our experiments show that CEM (1) attains better or competitive task accuracy w.r.t. standard DNNs trained without concept supervision, (2) learns concept representations that capture meaningful semantics at least as well as vanilla CBMs, and (3) supports effective test-time concept interventions.

# 2 Background

**Concept bottleneck models (CBMs, [9])**    A concept bottleneck model learns a mapping from samples $\mathbf{x} \in X$ to labels $y \in Y$ by means of: (i) a concept encoder function $g : X \to C$ which maps samples from the input space $\mathbf{x} \in X \subseteq \mathbb{R}^n$ to an intermediate space $\hat{\mathbf{c}} \in C \subseteq \mathbb{R}^k$ formed by $k$ concepts, and (ii) a label predictor function $f : C \to Y$ which maps samples from the concept space $\hat{\mathbf{c}} \in C$ to a downstream task space $\hat{\mathbf{y}} \in Y \subseteq \mathbb{R}^l$. A CBM requires a dataset composed of tuples in $X \times \mathcal{C} \times \mathcal{Y}$, where each sample consists of input features $\mathbf{x}$ (e.g., an image's pixels), ground truth concept vector $\mathbf{c} \in \{0, 1\}^k$ (i.e., a binary vector where each entry represents whether a concept is

active or not) and a task label $y$ (e.g., an image's class). During training, a CBM is encouraged to align $\hat{\mathbf{c}} = g(\mathbf{x})$ and $\hat{\mathbf{y}} = f(g(\mathbf{x}))$ to $\mathbf{x}$'s corresponding ground truth concepts $\mathbf{c}$ and task labels $y$, respectively. This can be done by (i) *sequentially* training first the concept encoder and then using its output to train the label predictor, (ii) *independently* training the concept encoder and label predictor and then combining them to form a CBM, or (iii) *jointly* training the concept encoder and label predictor via a weighted sum of cross entropy losses.

**Concept representations in CBMs**  For each sample $\mathbf{x} \in X$, the concept encoder $g$ learns $k$ different scalar concept representations $\hat{c}_1, \ldots, \hat{c}_k$. Boolean and Fuzzy CBMs [9] assume that each dimension of $\hat{\mathbf{c}}$, which we describe by $\hat{c}_i = s(\hat{\mathbf{c}})_{[i]} \in [0, 1]$, is aligned with a single ground truth concept and represents a probability of that concept being active. The element-wise activation function $s : \mathbb{R} \to [0, 1]$ can be either a thresholding function $s(x) \triangleq \mathbb{1}_{x \geq 0.5}$ (*Boolean* CBM) or sigmoidal function $s(x) \triangleq 1/(1 + e^{-x})$ (*Fuzzy* CBM).[2] A natural extension of this framework is a *Hybrid* CBM [14], where $\hat{\mathbf{c}} \in \mathbb{R}^{(k+\gamma)}$ contains $\gamma$ unsupervised dimensions and $k$ supervised concept dimensions which, when concatenated, form a shared concept vector (i.e., an "embedding").

**Concept Interventions in CBMs**  Interventions are one of the core motivations behind CBMs [9]. Through interventions, concept bottleneck models allow experts to improve a CBM's task performance by rectifying mispredicted concepts by setting, at test-time, $\hat{c}_i := c_i$ (where $c_i$ is the ground truth value of the $i$-th concept). Such interventions can significantly improve CBMs performance within a human-in-the-loop setting [9]. Furthermore, interventions enable the construction of meaningful concept-based counterfactuals [7]. For example, intervening on a CBM trained to predict bird types from images can determine that when the size of a "black" bird with "black" beak changes from "medium" to "large", while all other concepts remain constant, then one may classify the bird as a "raven" rather than a "crow".

## 3   Concept Embedding Models

In real-world settings, where complete concept annotations are costly and rare, vanilla CBMs may need to compromise their task performance in order to preserve their interpretability [9]. While Hybrid CBMs are able to overcome this issue by adding extra capacity in their bottlenecks, this comes at the cost of their interpretability and their responsiveness to concept interventions, thus undermining user trust [13]. To overcome these pitfalls, we propose *Concept Embedding Models* (CEMs), a concept-based architecture which represents each concept as a supervised vector. Intuitively, using high-dimensional embeddings to represent each concept allows for extra *supervised* learning capacity, as opposed to Hybrid models where the information flowing through their *unsupervised* bottleneck activations is concept-agnostic. In the following section, we introduce our architecture and describe how it learns a mixture of two semantic embeddings for each concept (Figure 2). We then discuss how interventions are performed in CEMs and introduce *RandInt*, a train-time regularisation mechanism that incentivises our model to positively react to interventions at test-time.

### 3.1   Architecture

For each concept, CEM learns a mixture of two embeddings with explicit semantics representing the concept's activity. Such design allows our model to construct evidence both in favour of and against a concept being active, and supports simple concept interventions as one can switch between the two embedding states at intervention time.

We represent concept $c_i$ with two embeddings $\hat{\mathbf{c}}_i^+, \hat{\mathbf{c}}_i^- \in \mathbb{R}^m$, each with a specific semantics: $\hat{\mathbf{c}}_i^+$ represents its active state (concept is `true`) while $\hat{\mathbf{c}}_i^-$ represents its inactive state (concept is `false`). To this aim, a DNN $\psi(\mathbf{x})$ learns a latent representation $\mathbf{h} \in \mathbb{R}^{n_{\text{hidden}}}$ which is the input to CEM's embedding generators. CEM then feeds $\mathbf{h}$ into two concept-specific fully connected layers, which learn two concept embeddings in $\mathbb{R}^m$, namely $\hat{\mathbf{c}}_i^+ = \phi_i^+(\mathbf{h}) = a(W_i^+ \mathbf{h} + \mathbf{b}_i^+)$ and $\hat{\mathbf{c}}_i^- = \phi_i^-(\mathbf{h}) = a(W_i^- \mathbf{h} + \mathbf{b}_i^-)$.[3] Notice that while more complicated models can be used to parameterise our concept embedding generators $\phi_i^+(\mathbf{h})$ and $\phi_i^-(\mathbf{h})$, we opted for a simple one-layer neural network to constrain

---

[2]In practice (e.g., [9]) one may use logits rather than sigmoidal activations to improve gradient flow [17].

[3]In practice, we use a leaky-ReLU for the activation $a(\cdot)$.

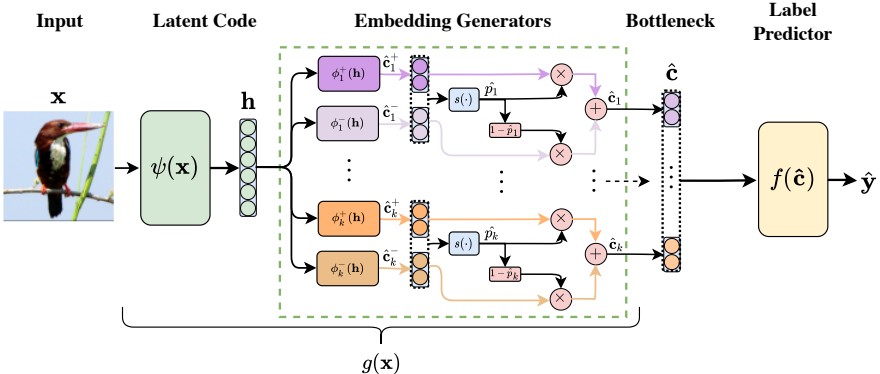

Figure 2: **Concept Embedding Model**: from an intermediate latent code $\mathbf{h}$, we learn two embeddings per concept, one for when it is active (i.e., $\hat{\mathbf{c}}_i^+$), and another when it is inactive (i.e., $\hat{\mathbf{c}}_i^-$). Each concept embedding (shown in this example as a vector with $m = 2$ activations) is then aligned to its corresponding ground truth concept through the scoring function $s(\cdot)$, which learns to assign activation probabilities $\hat{p}_i$ for each concept. These probabilities are used to output an embedding for each concept via a weighted mixture of each concept's positive and negative embedding.

parameter growth in models with large bottlenecks. Our architecture encourages embeddings $\hat{\mathbf{c}}_i^+$ and $\hat{\mathbf{c}}_i^-$ to be aligned with ground-truth concept $c_i$ via a learnable and differentiable scoring function $s : \mathbb{R}^{2m} \to [0, 1]$, trained to predict the probability $\hat{p}_i \triangleq s([\hat{\mathbf{c}}_i^+, \hat{\mathbf{c}}_i^-]^T) = \sigma\big(W_s[\hat{\mathbf{c}}_i^+, \hat{\mathbf{c}}_i^-]^T + \mathbf{b}_s\big)$ of concept $c_i$ being active from the embeddings' joint space. For the sake of parameter efficiency, parameters $W_s$ and $\mathbf{b}_s$ are shared across all concepts. Once both semantic embeddings are computed, we construct the final concept embedding $\hat{\mathbf{c}}_i$ for $c_i$ as a weighted mixture of $\hat{\mathbf{c}}_i^+$ and $\hat{\mathbf{c}}_i^-$:

$$\hat{\mathbf{c}}_i \triangleq \big(\hat{p}_i\hat{\mathbf{c}}_i^+ + (1 - \hat{p}_i)\hat{\mathbf{c}}_i^-\big)$$

Intuitively, this serves a two-fold purpose: (i) it forces the model to depend only on $\hat{\mathbf{c}}_i^+$ when the $i$-th concept is active, that is, $c_i = 1$ (and only on $\hat{\mathbf{c}}_i^-$ when inactive), leading to two different semantically meaningful latent spaces, and (ii) it enables a clear intervention strategy where one switches the embedding states when correcting a mispredicted concept, as discussed below. Finally, all $k$ mixed concept embeddings are concatenated, resulting in a bottleneck $g(\mathbf{x}) = \hat{\mathbf{c}}$ with $k \cdot m$ units (see end of Figure 2). This is passed to the label predictor $f$ to obtain a downstream task label. In practice, following Koh et al. [9], we use an interpretable label predictor $f$ parameterised by a simple linear layer, though more complex functions could be explored too. Notice that as in vanilla CBMs, CEM provides a concept-based explanation for the output of $f$ through its concept probability vector $\hat{\mathbf{p}}(\mathbf{x}) \triangleq [\hat{p}_1, \cdots, \hat{p}_k]$, indicating the predicted concept activity. This architecture can be trained in an end-to-end fashion by *jointly* minimising via stochastic gradient descent a weighted sum of the cross entropy loss on both task prediction and concept predictions:

$$\mathcal{L} \triangleq \mathbb{E}_{(\mathbf{x},y,\mathbf{c})}\Big[\mathcal{L}_{\text{task}}\big(y, f\big(g(\mathbf{x})\big)\big) + \alpha\mathcal{L}_{\text{CrossEntr}}\big(\mathbf{c}, \hat{\mathbf{p}}(\mathbf{x})\big)\Big] \tag{1}$$

where hyperparameter $\alpha \in \mathbb{R}^+$ controls the relative importance of concept and task accuracy.

### 3.2 Intervening with Concept Embeddings

As in vanilla CBMs, CEMs support test-time concept interventions. To intervene on concept $c_i$, one can update $\hat{\mathbf{c}}_i$ by swapping the output concept embedding for the one semantically aligned with the concept ground truth label. For instance, if for some sample $\mathbf{x}$ and concept $c_i$ a CEM predicted $\hat{p}_i = 0.1$ while a human expert knows that concept $c_i$ is active ($c_i = 1$), they can perform the intervention $\hat{p}_i := 1$. This operation updates CEM's bottleneck by setting $\hat{\mathbf{c}}_i$ to $\hat{\mathbf{c}}_i^+$ rather than $\big(0.1\hat{\mathbf{c}}_i^+ + 0.9\hat{\mathbf{c}}_i^-\big)$. Such an update allows the downstream label predictor to act on information related to the corrected concept. In addition, we introduce *RandInt*, a regularisation strategy exposing CEMs to concept interventions during training to improve the effectiveness of such actions at test-time. RandInt randomly performs independent concept interventions during training with probability $p_{\text{int}}$

(i.e., $\hat{p}_i$ is set to $\hat{p}_i := c_i$ for concept $c_i$ with probability $p_{\text{int}}$). In other words, for all concepts $c_i$, during training we compute embedding $\hat{\mathbf{c}}_i$ as:

$$\hat{\mathbf{c}}_i = \begin{cases} \left(c_i \hat{\mathbf{c}}_i^+ + (1 - c_i)\hat{\mathbf{c}}_i^-\right) & \text{with probability } p_{\text{int}} \\ \left(\hat{p}_i \hat{\mathbf{c}}_i^+ + (1 - \hat{p}_i)\hat{\mathbf{c}}_i^-\right) & \text{with probability } (1 - p_{\text{int}}) \end{cases}$$

while at test-time we always use the predicted probabilities for performing the mixing. During backpropagation, this strategy forces feedback from the downstream task to update only the correct concept embedding (e.g., $\hat{\mathbf{c}}_i^+$ if $c_i = 1$) while feedback from concept predictions updates both $\hat{\mathbf{c}}_i^+$ and $\hat{\mathbf{c}}_i^-$. Under this view, RandInt can be thought of as learning an average over an exponentially large family of CEM models (similarly to dropout [18]) where some of the concept representations are trained using only feedback from their concept label while others receive training feedback from both their concept and task labels.

## 4 Evaluating concept bottlenecks

To the best of our knowledge, while a great deal of attention has been paid to concept-based explainability in recent years, existing work still fails to provide methods that can be used to evaluate the interpretability of a concept embedding or to explain why certain CBMs underperform in their task predictions. With this in mind, we propose (i) a new metric for evaluating concept quality in multidimensional representations and (ii) an information-theoretic method which, by analysing the information flow in concept bottlenecks, can help understand why a CBM may underperform in a downstream task.

**Concept Alignment Score (CAS)**    The Concept Alignment Score (CAS) aims to measure how much learnt concept representations can be trusted as faithful representations of their ground truth concept labels. Intuitively, CAS generalises concept accuracy by considering the homogeneity of predicted concept labels within groups of samples which are close in a concept subspace. More specifically, for each concept $c_i$ the CAS applies a clustering algorithm $\kappa$ to find $\rho > 2$ clusters, assigning to each sample $\mathbf{x}^{(j)}$ a cluster label $\pi_i^{(j)} \in \{1, \cdots, \rho\}$. We compute this label by clustering samples using their $i$-th concept representations $\{\hat{\mathbf{c}}_i^{(1)}, \hat{\mathbf{c}}_i^{(2)}, \cdots\}$. Given $N$ test samples, the homogeneity score $h(\cdot)$ [19] then computes the conditional entropy $H$ of ground truth labels $C_i = \{c_i^{(j)}\}_{j=1}^N$ w.r.t. cluster labels $\Pi_i(\kappa, \rho) = \{\pi_i^{(j)}\}_{j=1}^N$, that is, $h = 1$ when $H(C_i, \Pi_i) = 0$ and $h = 1 - H(C_i, \Pi_i)/H(C_i)$ otherwise. The higher the homogeneity, the more a learnt concept representation is "aligned" with its labels, and can thus be trusted as a faithful representation. CAS averages homogeneity scores over all concepts and number of clusters $\rho$, providing a normalised score in $[0, 1]$:

$$\text{CAS}(\hat{\mathbf{c}}_1, \cdots, \hat{\mathbf{c}}_k) \triangleq \frac{1}{N - 2} \sum_{\rho=2}^{N} \left( \frac{1}{k} \sum_{i=1}^{k} h(C_i, \Pi_i(\kappa, \rho)) \right) \tag{2}$$

To tractably compute CAS in practice, we sum homogeneity scores by varying $\rho$ across $\rho \in \{2, 2 + \delta, 2 + 2\delta, \cdots, N\}$ for some $\delta > 1$ (details in Appendix A.1). Furthermore, we use k-Medoids [20] for cluster discovery, as used in Ghorbani et al. [10] and Magister et al. [21], and use concept logits when computing the CAS for Boolean and Fuzzy CBMs. For Hybrid CBMs, we use $\hat{\mathbf{c}}_i \triangleq [\hat{\mathbf{c}}_{[k:k+\gamma]}, \hat{\mathbf{c}}_{[i:(i+1)]}]^T$ as the concept representation for $c_i$ given that the extra capacity is a shared embedding across all concepts.

**Information bottleneck**    The relationship between the quality of concept representations w.r.t. the input distribution remains widely unexplored. Here we propose to analyse this relationship using information theory methods for DNNs developed in Tishby et al. [22] and Tishby and Zaslavsky [23]. In particular, we compare concept bottlenecks using the Information Plane method [22] to study the information flow at concept level. To this end, we measure the evolution of the Mutual Information ($I(\cdot, \cdot)$) of concept representations w.r.t. the input and output distributions across training epochs. We conjecture that embedding-based CBMs circumvent the information bottleneck by preserving more information than vanilla CBMs from the input distribution as part of their high-dimensional activations. If true, such effect should be captured by Information Planes in the form of a positively correlated evolution of $I(X, \hat{C})$, the Mutual Information (MI) between inputs $X$ and learnt concept

representations $\hat{C}$, and $I(\hat{C}, Y)$, the MI between learnt concept representations $\hat{C}$ and task labels $Y$. In contrast, we anticipate that scalar-based concept representations (e.g., Fuzzy and Bool CBMs), will be forced to compress the information from the input data at concept level, leading to a compromise between the $I(X, \hat{C})$ and $I(\hat{C}, Y)$. Further details on our implementation are in Appendix A.2.

## 5  Experiments

In this section, we address the following research questions:

- **Task accuracy** — What is the impact of concept embeddings on a CBM's downstream task performance? Are models based on concept embeddings still subject to an information bottleneck [22]?
- **Interpretability** — Are CEM concept-based explanations aligned with ground truth concepts? Do they offer interpretability beyond simple concept prediction and alignment?
- **Interventions** — Do CEMs allow meaningful concept interventions when compared to Hybrid or vanilla CBMs?

### 5.1  Setup

**Datasets**  For our evaluation, we propose three simple benchmark datasets of increasing concept complexity (from Boolean to vector-based concepts): (1) *XOR* (inspired by [24]) in which tuples $(x_1, x_2) \in [0, 1]^2$ are annotated with two Boolean concepts $\{\mathbb{1}_{c_i > 0.5}\}_{i=1}^2$ and labeled as $y = c_1$ XOR $c_2$; (2) *Trigonometric* (inspired by [14]) in which three latent normal random variables $\{b_i\}_{i=1}^3$ are used to generate a 7-dimensional input whose three concept annotations are a Boolean function of $\{b_i\}_{i=1}^3$ and task label is a linear function of the same; (3) *Dot* in which four latent random vectors $\mathbf{v}_1, \mathbf{v}_2, \mathbf{w}_1, \mathbf{w}_2 \in \mathbb{R}^2$ are used to generate two concept annotations, representing whether latent vectors $\mathbf{v}_i$ point in the same direction of reference vectors $\mathbf{w}_i$, and task labels, representing whether the two latent vectors $\mathbf{v}_1$ and $\mathbf{v}_2$ point in the same direction. Furthermore, we evaluate our methods on two real-world image tasks: the Caltech-UCSD Birds-200-2011 dataset (CUB, [16]), preprocessed as in [9], and the Large-scale CelebFaces Attributes dataset (CelebA, [25]). In our CUB task we have 112 complete concept annotations and 200 task labels while in our CelebA task we construct 6 balanced incomplete concept annotations and each image can be one of 256 classes. Therefore, we use CUB to test each model in a real-world task where concept annotations are numerous and they form a complete description of their downstream task. In contrast, our CelebA task is used to evaluate the behaviour of each method in scenarios where the concept annotations are scarce and incomplete w.r.t. their downstream task. Further details on these datasets and their properties are provided in Appendix A.3.

**Baselines**  We compare CEMs against Bool, Fuzzy, and Hybrid Joint-CBMs as they all provide concept-based explanations for their predictions and allow concept interventions at test-time. Note that this set excludes architectures such as Self-Explainable Neural Networks [26] and Concept Whitening [12] as they do not offer a clear mechanism for intervening on their concept bottlenecks. To ensure fair comparison, we use the same architecture capacity across all models . We empirically justify this decision in Appendix A.4 by showing that the results discussed in this section do not change if one modifies the underlying model architecture. Similarly, we use the same values of $\alpha$ and $m$ within a dataset for all models trained on that dataset and set $p_{\text{int}} = 0.25$ when using CEM (see Appendix A.5 for an ablation study of $p_{\text{int}}$ showing how intervention improvement plateaus around this value). When using Hybrid CBMs, we include as many activations in their bottlenecks as their CEM counterparts (so that they both end up with a bottleneck with $km$ activations) and use a Leaky-ReLU activation for unsupervised activations. Finally, in our evaluation we include a DNN without concept supervision with the same capacity as its CEM counterpart to measure the effect of concept supervision in our model's performance. For further details on the model architectures and training hyperparameters, please refer to Appendix A.6.

**Metrics**  We measure a model's performance based on four metrics. First, we measure task and concept classification performance in terms of both *task and mean concept accuracy*. Second, we evaluate the interpretability of learnt concept representations via our *concept alignment score*. To easily visualise the accuracy-vs-interpretability trade-off, we plot our results in a two-dimensional

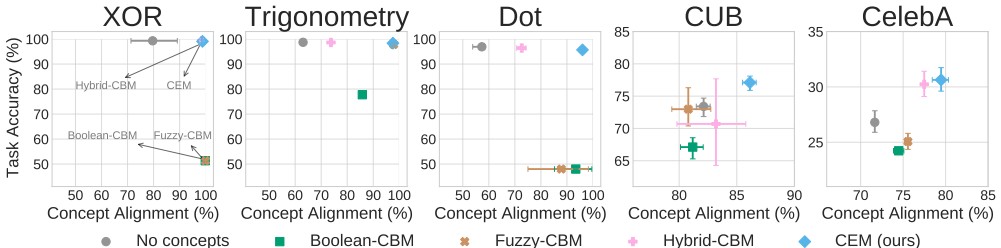

Figure 3: Accuracy-vs-interpretability trade-off in terms of **task accuracy** and **concept alignment score** for different concept bottleneck models. In CelebA, our most constrained task, we show the top-1 accuracy for consistency with other datasets.

plane showing both task accuracy and concept alignment. Third, we study the information bottleneck in our models via *mutual information* (MI) and the Information Plane technique [27]. Finally, we quantify user trust [13] by evaluating a model's task performance after concept interventions. All metrics in our evaluation, across all experiments, are computed on test sets using 5 random seeds, from which we compute a metric's mean and $95\%$ confidence interval using the Box-Cox transformation for non-normal distributions.

## 5.2 Task Accuracy

**CEM improves generalisation accuracy (y-axis of Figure 3)**    Our evaluation shows that embedding-based CBMs (i.e., Hybrid-CBM and CEM) can achieve competitive or better downstream accuracy than DNNs that do not provide any form of concept-based explanations, and can easily outperform Boolean and Fuzzy CBMs by a large margin (up to $+45\%$ on Dot). This effect is emphasised when the downstream task is not a linear function of the concepts (e.g., XOR and Trigonometry) or when concept annotations are incomplete (e.g., Dot and CelebA). At the same time, we observe that all concept-based models achieve a similar high mean concept accuracy across all datasets (see Appendix A.7). This suggests that, as hypothesised, the trade-off between concept accuracy and task performance in concept-incomplete tasks is significantly alleviated by the introduction of concept embeddings in a CBM's bottleneck. Similar results can be observed when training our baselines using only a fraction of the available concepts in CUB as seen in Appendix A.8. Finally, notice that CelebA showcases how including concept supervision during training (as in CEM) can lead to an even higher task accuracy than the one obtained by a vanilla end-to-end model ($+5\%$ compared to "No concepts" model). This result further suggests that concept embedding representations enable high levels of interpretability without sacrificing performance.

**CEM overcomes the information bottleneck (Figure 4)**    The Information Plane method indicates, as hypothesised, that embedding-based CBMs (i.e., Hybrid-CBM and CEM) do not compress input data information, with $I(X, \hat{C})$ monotonically increasing during training epochs. On the other hand, Boolean and Fuzzy CBMs, as well as vanilla end-to-end models, tend to "forget" [27] input data information in their attempt to balance competing objective functions. Such a result constitutes a plausible explanation as to why embedding-based representations are able to maintain both high task accuracy and mean concept accuracy compared to CBMs with scalar concept representations. In fact, the extra capacity allows CBMs to maximise concept accuracy without over-constraining concept representations, thus allowing useful input information to pass by. In CEMs all input information flows through concepts, as they supervise the whole concept embedding. In contrast with Hybrid models, this makes the downstream tasks completely dependent on concepts, which explains the higher concept alignment scores obtained by CEM (as discussed in the next subsection).

## 5.3 Interpretability

**CEM learns more interpretable concept representations (x-axis of Figure 3)**    Using the proposed CAS metric, we show that concept representations learnt by CEMs have alignment scores competitive or even better (e.g., on CelebA) than the ones of Boolean and Fuzzy CBMs. The alignment score also shows, as hypothesised, that hybrid concept embeddings are the least faithful representations—with

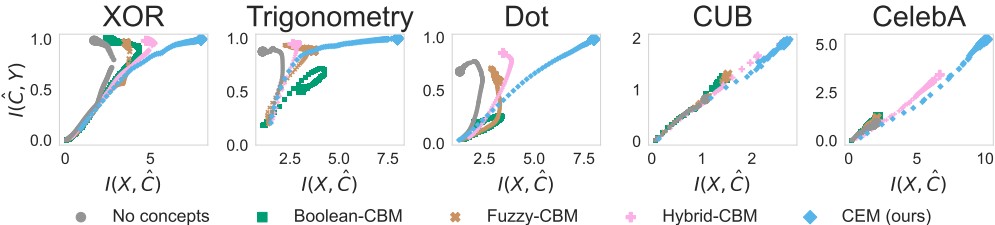

Figure 4: Mutual Information (MI) of concept representations ($\hat{C}$) w.r.t. input distribution ($X$) and ground truth labels ($Y$) during training. The size of the points is proportional to the training epoch.

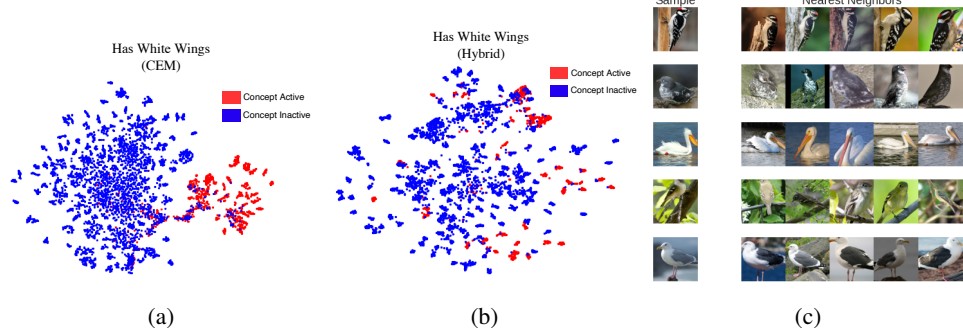

(a)                                    (b)                                    (c)

Figure 5: Qualitative results: (a and b) t-SNE visualisations of "has white wings" concept embedding learnt in CUB with sample points coloured red if the concept is active in that sample, (c) top-5 test neighbours of CEM's embedding for the concept "has white wings" across 5 random test samples.

alignment scores up to $25\%$ lower than CEM in the Dot dataset. This is due to their unsupervised activations containing information which may not be necessarily relevant to a given concept. This result is a further evidence for why we expect interventions to be ineffective in Hybrid models (as we show shortly).

**CEM captures meaningful concept semantics (Figure 5)** Our concept alignment results hint at the possibility that concept embeddings learnt by CEM may be able to offer more than simple concept prediction. In fact, we hypothesise that their seemingly high alignment may lead to these embeddings forming more interpretable representations than Hybrid embeddings, which can lead to these embeddings serving as better representations for different tasks. To explore this, we train a Hybrid-CBM and a CEM, both with $m = 16$, using a variation of CUB with only $25\%$ of its concept annotations randomly selected before training, resulting in a bottleneck with 28 concepts (see Appendix A.9 for details). Once these models have been trained to convergence, we use their learnt bottleneck representations to predict the remaining $75\%$ of the concept annotations in CUB using a simple logistic linear model. The model trained using the Hybrid bottleneck notably underperfoms when compared to the model trained using the CEM bottleneck (Hybrid-trained model has a mean concept accuracy of $91.83\% \pm 0.51\%$ while the CEM-trained model's concept accuracy is $94.33\% \pm 0.88\%$). This corroborates our CAS results by suggesting that the bottlenecks learnt by CEMs are considerably more interpretable and can therefore serve as powerful feature extractors.

We can further explore this phenomena qualitatively by visualising the embeddings learnt for a single concept using its 2-dimensional t-SNE [28] plot. As shown in colour in Figure 5a, we can see that the embedding space learnt for a concept $\hat{c}_i$ (here we show the concept "has white wings") forms two clear clusters of samples, one for points in which the concept is active and one for points in which the concept is inactive. When performing a similar analysis for the same concept in the Hybrid CBM (Figure 5b), where we use the entire extra capacity $\hat{c}_{[k:k+\gamma]}$ as the concept's embedding representation, we see that this latent space is not as clearly separable as that in CEM's embeddings, suggesting this latent space is unable to capture concept-semantics as clearly as CEM's latent space. Notice that CEM's t-SNE seems to also show smaller subclusters within the activated and inactivated clusters. As Figure 5c shows, by looking at the nearest Euclidean neighbours in concept's $c_i$ embedding's space,

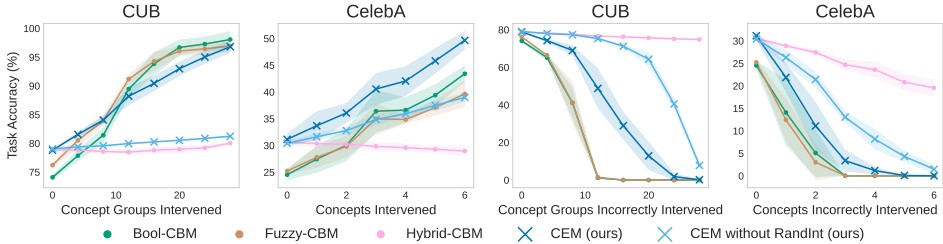

Figure 6: Effects of performing positive random concept interventions (left and center left) and incorrect random interventions (center right and right) for different models in CUB and CelebA. As in [9], when intervening in CUB we jointly set groups of mutually exclusive concepts.

we see that CEM concepts clearly capture a concept's activation, as well as exhibit high class-wise coherence by mapping same-type birds close to each other (explaining the observed subclusters). These results, and similar qualitative results in Appendix A.10, suggest that CEM is learning a hierarchy in its latent space where embeddings are separated with respect to their concept activation and, within the set of embeddings that have the same activation, embeddings are clustered according to their task label.

## 5.4 Interventions

**CEM supports effective concept interventions and is more robust to incorrect interventions (Figure 6)** When describing our CEM architecture, we argued in favour of using a mixture of two semantic embeddings for each concept as this would permit test-time interventions which can meaningfully affect entire concept embeddings. In Figure 6 left and center-left, we observe, as hypothesised, that using a mixture of embeddings allows CEMs to be highly responsive to random concept interventions in their bottlenecks. Notice that although all models have a similar concept accuracy (see Appendix A.7), we observe that Hybrid CBMs, while highly accurate without interventions, quickly fall short against even scalar-based CBMs once several concepts are intervened in their bottlenecks. In fact, we observe that interventions in Hybrid CBM bottlenecks have little effect on their predictive accuracy, something that did not change if logit concept probabilities were used instead of sigmoidal probabilities. However, even interventions performed by human experts are quite rarely perfect. For this reason, we simulate incorrect interventions (where a concept is set to the wrong value) to measure how robust the model is to such errors. We observe (Figure 6 center-right and right) that CEM's performance deteriorates as more concepts are incorrectly intervened on (as opposed to hybrid-CBMs), while it can withstand a few errors without losing much performance (as opposed to Bool and Fuzzy-CBMs). We suggest that this is a consequence of CEM's "incorrect" embeddings still carrying important task-specific information which can then be used by the label predictor to produce more accurate task labels, something worth exploring in future work. As a result, users can trust CEMs to better handle a small number of accidental mistakes made by human experts when intervening in its concept activations. Finally, by comparing the effect of interventions in both CEMs and CEMs trained without RandInt, we observe that RandInt in fact leads to a model that is not just significantly more receptive to interventions, but is also able to outperform even scalar-based CBMs when large portions of their bottleneck are artificially set by experts (e.g., as in CelebA). This, as shown in Appendix A.11, comes without a significant computational training costs for CEM. Interestingly, such a positive effect in concept interventions is not observed if RandInt is used when training our other baselines (see Appendix A.12 for an explanation). This suggests that our proposed architecture can not only be trusted in terms of its downstream predictions and concept explanations, as seen above, but it can also be a highly effective model when used along with experts that can correct mistakes in their concept predictions. For further details, including an exploration of performing interventions with Sequential and Independent CBMs, please refer to Appendix A.13.

## 6 Discussion

**Relations with the state-of-the-art** Concept bottleneck models engender user trust [13] by (i) forcing information flow through concept-aligned activations during training and (ii) by supporting

human interventions in their concept predictions. This allows CBMs to circumvent the well-known unreliability of post-hoc methods [29, 30, 10], such as saliency maps [31–33], and invites their use in settings where input features are naturally hard to reason about (e.g., raw image pixels). In addition, CBMs encourage human interactions allowing experts to improve task performance by rectifying mispredicted concepts, which contrasts other concept-based interpretable architectures such as Self-Explainable Neural Networks [26] and Concept Whitening [12]. However, our experiments show that all existing CBMs are limited to significant accuracy-vs-interpretability trade-offs. In this respect, our work reconciles theoretical results with empirical observations: while theoretical results suggest that explicit per-concept supervisions should improve generalisation bounds [34], in contrast Koh et al. [9], Chen et al. [12], and Mahinpei et al. [14] empirically show how learning with intermediate concepts may impair task performance in practice. The Information Plane method [27] reveals that the higher generalisation error of existing concept bottleneck models might be explained as a compression in the input information flow caused by narrow architectures of Boolean and Fuzzy CBMs. In contrast, CEM represents the first concept-based model which does not need to compromise between task accuracy, concept interpretability or intervention power, thus filling this gap in the literature. Furthermore, through an ablation study on CEM's embedding size shown in Appendix A.14, we see that one does not need to increase the embedding size drastically to begin to see the benefits of using a CEM over a vanilla CBM or an end-to-end black box DNN. We note that as stronger interpretable models are deployed, there are risks of societal harm which we must be vigilant to avoid.

**Conclusion** Our experiments provide significant evidence in favour of CEM's accuracy/interpretability and, consequently, in favour of its real-world deployment. In particular, CEMs offer: (i) state-of-the-art task accuracy, (ii) interpretable concept representations aligned with human ground truths, (iii) effective interventions on learnt concepts, and (iv) robustness to incorrect concept interventions. While in practice CBMs require carefully selected concept annotations during training, which can be as expensive as task labels to obtain, our results suggest that CEM is more efficient in concept-incomplete settings, requiring less concept annotations and being more applicable to real-world tasks. While there is room for improvement in both concept alignment and task accuracy in challenging benchmarks such as CUB or CelebA, as well as in resource utilisation during inference/training (see Appendix A.11), our results indicate that CEM advances the state-of-the-art for the accuracy-vs-interpretability trade-off, making progress on a crucial concern in explainable AI.

## Acknowledgments and Disclosure of Funding

The authors would like to thank Carl Henrik Ek, Alberto Tonda, Andrei Margeloiu, Fabrizio Silvestri, and Maria Sofia Bucarelli for their insightful comments on earlier versions of this manuscript. MEZ acknowledges support from the Gates Cambridge Trust via a Gates Cambridge Scholarship. PB acknowledges support from the European Union's Horizon 2020 research and innovation programme under grant agreement No 848077. GC and FP acknowledges support from the EU Horizon 2020 project AI4Media, under contract no. 951911 and by the French government, through Investments in the Future projects managed by the National Research Agency (ANR), 3IA Cote d'Azur with the reference number ANR-19-P3IA-0002. AW acknowledges support from a Turing AI Fellowship under grant EP/V025279/1, The Alan Turing Institute, and the Leverhulme Trust via CFI. GM is funded by the Research Foundation-Flanders (FWO-Vlaanderen, GA No 1239422N). FG is supported by TAILOR, a project funded by EU Horizon 2020 research and innovation programme under GA No 952215. This work was also partially supported by HumanE-AI-Net a project funded by EU Horizon 2020 research and innovation programme under GA 952026. MJ is supported by the EPSRC grant EP/T019603/1.

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
