# OpenReview forum: "Concept Embedding Models: Beyond the Accuracy-Explainability Trade-Off"
_NeurIPS.cc/2022/Conference — NeurIPS 2022 Accept_

### Official Review · Reviewer_6on2 · 2022-07-09

**Rating:** 5
**Confidence:** 4
**Soundness:** 3 good
**Presentation:** 3 good
**Contribution:** 2 fair

**Summary:**

This paper introduces a model named concept embedding model (CEM) based on concept bottleneck model (CBM) architecture. Compared to CBM, CEM contains an embedding generator layer that considers two embedding representations (one for activate and one for inactivate) and then produces an embedding representation for one concept. Results show that the model produces high task accuracy and interpretability at the same time compared to CBM-family models.

**Questions:**

(1)	Line 163-164, p should be rho?

(2)	Sec 5.4, line 312-314 suggests that CEM is robust to incorrect concepts in intervention. But is it desirable in the practice? Do authors consider the scenario where changing the important concept that can change the final decision?  If so, how it can truly reflex the user trust?


**Limitations:**

This paper does not emphasize the advantage of using two separate concept representations (c_hat+ and c_hat-), which is the novelty of the CEM, and does not evaluate the interpretability and user trust thoroughly.

**Strengths And Weaknesses:**

Strength: This paper tries to solve a challenging research question: to design XAI models which are good at task performance and interpretability. The authors conduct experiments on multiple datasets and evaluate different models using different metrics.

Weakness:
(1) The proposed model is not thoroughly studied. For instance, the relationship between c_hat+ and c_hat−. This is the novelty of the proposed model compared to CBM. For instance, for one concept, do these two embeddings represent opposite concepts?

(2) There is some unclearness about the baseline models in the paper. For example, the CEM uses m=16 to represent one concept in c_hat (bottleneck). What is the dimension of c_hat for CBMs? In Appendix A.5, it says “γ = k · (m − 1)” for Hybrid-CBM. Does it mean that the dimension of c_hat is k · (m − 1) dimension? Why not k*m as in CEM?

(3) The lack of justification for the proposed CAS. In Fig.3, the baseline model “no concepts” has an even better score than Boolean-CBM and Fuzzy-CBM.

(4) The qualitative results are not very convincing. Fig.5 c and Appendix Fig. 5 show the samples and their nearest neighbors for one concept. However, it does not reflex information about the concept and for a well-trained classifier, it should find out the visually similar samples based on Euclidean distance in the embedding space.

---

> ### Author Response · Authors · 2022-08-02
> **Answers to Rev-6on2 (2)**
>
> **Takeaways from qualitative results:** the purpose of our qualitative experiments is to showcase some interesting properties that emerge in our method’s embeddings. For this, we first show that the embedding spaces for both $\mathbf{\hat{c}}^+$ and $\mathbf{\hat{c}}^-$ are highly disjoint (Fig 5(a)). Then we show in Fig 5(c) that the overall generated concept embedding can lead to a highly class-coherent latent space (both when the concept is ON and OFF). Together, these results suggest that **CEM is learning a hierarchy** in its latent space where embeddings are separated with respect to their concept activation and within the set of embeddings that have the same activation, embeddings are highly grouped according to their downstream task label. We updated section 5.3 to emphasize this.
>
> **Lack of proper interpretability and user trust evaluation:** Given the lack of agreed-upon interpretability metrics, in this work we evaluate the interpretability of our concept representations using our novel CAS score and our model’s concept accuracy. Moreover, while we believe evaluating user trust through formal user studies can be interesting future work, given our paper’s already qualitative and quantitative experiments, we opted to instead **evaluate the efficiency of concept interventions as they are strongly related to user trust [1] and were used in the original CBM paper for this very objective [2]**.
>
> [1] Shen, Max W. "Trust in AI: Interpretability is not necessary or sufficient, while black-box interaction is necessary and sufficient." arXiv:2202.05302 (2022).
>
> [2] Koh, Pang Wei, et al. "Concept bottleneck models." International Conference on Machine Learning. PMLR, 2020.

---

> > ### Comment · Reviewer_6on2 · 2022-08-08
> > **Reply to the rebuttal**
> >
> > Dear authors,
> >
> > thank you for your reply and explanations, especially since it solves some of my misunderstandings about the c_hat+ and c_hat-. The proposed CEM seems to work surprisingly well with a simple minor change in the architecture, compared to CBM. I upgrade my score by one level to show that I acknowledge the rebuttal.
> >
> > Moreover, I still have some concerns at this stage. I see the point in the robustness of the CEM when users make mistakes unintentionally. But, it may be worth studying, if users change a very critical/important attribute for that class. The model should change the prediction but not simply hold the same prediction.

---

> > > ### Author Response · Authors · 2022-08-09
> > > **Thank you for your response**
> > >
> > > Dear Reviewer 6on2,
> > >
> > > We would like to thank you for taking the time to respond to our rebuttal and for updating your score. We are happy to hear that our rebuttal and updated submission were able to address some of your concerns. Furthermore, we appreciate your very insightful feedback and look forward to exploring CEM's robustness to mistakes in more depth as part of follow-up work.

---

> ### Author Response · Authors · 2022-08-02
> **Answers to Rev-6on2 (1)**
>
> **Clarity of baseline architectures and dimensionality of $\mathbf{\hat{c}}$:** Given $k$ training concepts, for Fuzzy and Boolean CBMs the dimensionality of $\mathbf{\hat{c}}$ is $k$ (i.e., one activation per concept). In contrast, CEM's bottleneck $\mathbf{\hat{c}}$ has size $m \cdot k$, where we fix $m$ to 16.
> To ensure a fair comparison between CEM and Hybrid-CBM, we allow the number of extra activations of Hybrid-CBM to be $\gamma = k (m - 1)$. **This is so that the overall bottleneck of Hybrid-CBM has size $k + \gamma = k + k (m - 1) = k \cdot m$, as in our CEM model!** Notice that we add $k$ to $\gamma$ to compute the size of $\mathbf{\hat{c}}$ for Hybrid-CBM as this model has one activation for each concept we provide supervision for (i.e., $k$) plus some extra capacity $\gamma$. Thank you for pointing out this lack of clarity; we have updated Section 5.1 and App A.4 to better explain this.
>
> **Robustness of concepts and desirability of robustness to incorrect concept interventions:** These are excellent questions! Due to the lack of agreed-upon metrics for evaluating such concept robustness, we measured this property via human concept interventions, as interventions are the main advantage of CBMs over standard architectures. However, even interventions performed by human experts are quite rarely perfect. Hence, we simulate incorrect interventions by human experts to measure robustness in terms of how robust the model is to such errors:  it is desirable for the model to have some self-correction mechanism to withstand a few errors without losing much task performance (i.e., incorrectly changing a few concepts does not change the final decision), while the model task accuracy should drop significantly when many errors occur (after all, the model should depend on the concept settings). In Fig 6 we observe that CEM's performance in fact deteriorates as more concepts are incorrectly intervened on (as opposed to hybrid models), while it can withstand a few errors without losing much performance (as opposed to standard concept bottleneck models). **This suggests that a user can trust our approach to better handle a small number of accidental mistakes made by a human expert when intervening in its concept activations.** We thank the reviewers for their insightful comments and we include comments on this point in section 5.4.
>
> **The relationship between $\mathbf{\hat{c}}^+$ and $\mathbf{\hat{c}}^-$ is not well studied:** As indicated both by your comment and our description of CEM, $\mathbf{\hat{c}}^+$ and $\mathbf{\hat{c}}^-$ are evidence for a concept being activated or inactive, respectively(i.e., concept ON vs OFF). **We explored the relationship between these two embeddings both quantitatively and qualitatively.** We quantitatively evaluate the impact of this architectural novelty in terms of (1) the higher information flow through the concept bottleneck (MI and performance experiments) and (2) the improved effect of human interventions, and CEM’s robustness to intervention mistakes, due to our dual concept embedding design (Fig 6). Qualitatively, we explore the latent spaces formed by $\mathbf{\hat{c}}^+$ and $\mathbf{\hat{c}}^-$ and show that they form very distinct and highly separable spaces: Fig 5 (left) shows that our method separates the 2 classes well, whereas Fig 5 (center) shows the earlier hybrid method does not.
>
> **The lack of justification for CAS:** We propose the CAS to **address the lack of interpretability metrics applicable to concept embeddings**. Intuitively, the CAS approaches this by generalizing the concept predictive accuracy to embeddings. If a concept representation is able to capture a concept correctly, then we would expect that clustering samples based on that representation would result in coherent clusters where samples within the same cluster all have the concept active or inactive. The CAS captures this by looking at how coherent clusters are for each concept representation w.r.t. concept labels as we change the size of each cluster. We incorporate your useful feedback by better justifying this metric in App A.1.
>
> **CAS results for black-box DNN baseline:** Overall Fig 3 shows that **the baseline model “No Concepts” does not have a better CAS score than Boolean- or Fuzzy-CBMs for 4 out of 5 datasets**. The only case where this is actually true is for the CUB dataset and it does not appear to be statistically significant. Nevertheless, we have updated App A.7 to include a brief possible explanation for this.

---

### Official Review · Reviewer_GkfX · 2022-07-10

**Rating:** 7
**Confidence:** 4
**Soundness:** 3 good
**Presentation:** 3 good
**Contribution:** 3 good

**Summary:**

Concept bottleneck models implicitly learn to explain the downstream tasks in addition to learning how to perform them. However, these models forgo predictive performance on the downstream tasks. The authors propose Concept Embedding Models (CEMs), a novel family of concept bottleneck models that address this issue.

**Questions:**

- How robust are the concepts learned? Are they robust to changes in backgrounds?
- Is it possible to relax assumptions that datasets should contain concept annotations? Maybe train the model with a contrastive loss for concepts instead of a supervised loss.
- How does the underlying architecture of the model change the concepts generated?
- How much is the additional computational overhead incurred while training CEMs as opposed to CBMs?

I would encourage the authors to include results on OAI & AwA2 too.

**Limitations:**

I would encourage the authors to list the limitations of the proposed approach.

**Strengths And Weaknesses:**

Strengths:
- The proposed techniques perform better than CBMs and their variants on downstream tasks.
- Concept representation learned by CEMs is more aligned to the ground truth concepts and successfully captures the semantics of the images.
Weaknesses:
- CEMs assume that the datasets contain annotations of concepts which is not valid in practice and are often quite expensive to obtain.

---

> ### Author Response · Authors · 2022-08-02
> **Answers to Rev-GkfX**
>
> **CEM’s need for concept annotations and its cost:** All concept bottleneck models require datasets containing concept annotations. However, **our work relaxes this assumption by allowing concept annotations that are incomplete w.r.t. the downstream task to be used**. This allows a dramatic reduction of required concept annotations and the costs related to acquiring such annotations without the need for sacrificing performance or interpretability (as seen in our CelebA results in Fig 3). We furthermore highlight this property by showing in our new App A.10 (specifically in Fig A.6 of that App) that when one trains our baselines using only a subsample of the concepts available in CUB, CEM’s performance is significantly higher than that of CBMs as the number of concept annotations used is drastically decreased. Notice that while Hybrid-CBM is able to also maintain a high accuracy in concept-incomplete settings, this comes at the crucial cost of concept interventions (as shown in section 5.4).
>
> **Robustness of concept representations to background changes & use of contrastive loss to avoid the need for concept annotations:** These are both extremely interesting ideas that would be worth exploring as future work. Nevertheless, for the purposes of this paper we believe that they may be out-of-scope in this submission given the amount of experiments and contributions we already include in our evaluation.
>
>
> **Effect of underlying architecture in learnt concepts:** Great question! As expected, different architectures will have different approximation capabilities, and the resulting representations will be affected by the architectural choices. Still, for the sake of fairness in our evaluation we focused on using the same model architecture across all baselines, with the backbone in real-world datasets (CUB and CelebA) being a commonly used pretrained ResNet-34 model. Nevertheless, to explore whether this decision might’ve biased our results, we update Section 5.1 with a pointer to our new App A.9 where we show that the relative rankings across methods in our real-world tasks are preserved when using backbones with significantly different capacities (i.e., a ResNet18 vs a ResNet34). Finally, we would like to highlight that in App A.15 (part of our original submission) we also include an ablation of the embedding size, which is itself an architectural component. In these experiments we observe that unless the embedding size is highly constrained, our model is able to maintain a high accuracy while being highly receptive to interventions.
>
> **Results on OAI & AwA2:** We agree that this would certainly be interesting and useful. Nevertheless, we believe that our experiment section, which already includes **two real-world complex datasets and three synthetic ones**, provides a broad and meaningful evaluation of our proposed methodology. However, we would love to explore such datasets (and others) in future work that expands on our architecture.
>
> **Listing the limitations of our work:** We mention the limitations of our model in our Conclusion.

---

> > ### Comment · Reviewer_GkfX · 2022-08-06
> > **Response to Rebuttal**
> >
> > Dear Authors,
> >
> > Thanks for addressing/answering all my questions, and I am satisfied with the responses. I have updated my scores to reflect the same.

---

> > > ### Author Response · Authors · 2022-08-09
> > > **Thank you for your response**
> > >
> > > Dear Reviewer GkfX,
> > >
> > >
> > > We would like to thank you for taking the time to respond to our rebuttal and for updating your score. We are glad to hear that our answers and update have satisfied your questions and concerns regarding our work.

---

### Official Review · Reviewer_JCLo · 2022-07-17

**Rating:** 6
**Confidence:** 3
**Soundness:** 3 good
**Presentation:** 3 good
**Contribution:** 3 good

**Summary:**

The paper extends the concept bottleneck method by generating two embedding vectors for each concept; one representing the embedding when the concept is active while the other representing when the concept is inactive. These two embedding are linearly combined through a scoring function (similar to the gating mechanism) the produces a probability score of which embedding is to be used. The concept embedding vectors are then used in the downstream task for prediction. This extension increase the model capacity to encode more information about concepts.


**Questions:**

Read my listed points in the weakness section.

**Limitations:**

No.

**Strengths And Weaknesses:**

Strength
- easy to read
- simple extension

Weakness
- constraining the architecture of all other baselines as similar to the architecture of the proposed method seems to be unfair.
- I still can not see how this model can be used in scenarios where incomplete concepts exist. In particular, in eq 1, how can the model be trained without a full supervision of concepts?
- the information bottleneck metric in section 4 is a bit unclear. more detailed explanation would be preferred.

---

> ### Author Response · Authors · 2022-08-02
> **Answers to Rev-JCLo**
>
> **Fixing architecture across all baselines:** This is an important point to raise, thank you. In fact we consistently used the **same backbone architecture for all models** following similar architectures in the literature (i.e., a commonly used ResNet38), hence this should not hurt them in comparison. The purpose of constraining the architecture across all models is two-fold: (1) we simulate a real-world use of our methodology by focusing our evaluation on an out-of-the-box standard model (e.g., a commonly used ResNet38 as a backbone) and (2) we make sure that we fairly compare all methods, and avoid possible external confounding factors, by providing, when possible, each model with **the exact same computational capacity**. This means that the Hybrid-CBM and blackbox DNN models will have the exact same capacity, backbone, and label predictor as that of their contrasting CEM model, leading to a fair comparison across competing models. Furthermore, as indicated in our updated Section 5.1 and as seen in our results in Figs A.4 and A.5 of App A.9 (showing the effect of architecture choice in our results), the relative rankings observed across methods for predictive performance (with and without concept interventions) is maintained across methods even when the architecture is changed.
>
> **How can this model be used in scenarios where complete concepts are unavailable:** Thank you for raising this point, this is a great question whose answer is crucial for our work. In this paper, we use "concept incompleteness" to refer to a situation in which the concept annotations are "incomplete" w.r.t. a task of interest. As a simple example inspired by our CUB task, consider a dataset of labeled bird images where each image has only the concept annotations "red chest" and "white chest". Clearly, knowing the bird's chest color is typically not enough to predict the bird type, and therefore the given set of concept annotations is "incomplete" w.r.t. the task of predicting the bird type. In this example, when training a CEM using Equation 1, $\mathbf{c}$ will be a vector in $[0, 1]^2$ indicating the activation of each of our two concepts while every other concept that was not provided as part of our training data (e.g., “gray neck”) is not included in $\mathbf{c}$. What we highlight in this paper (see Fig 1 (b)) is that **current supervised concept-learning methods struggle** to find a good balance between interpretability and task accuracy **when the set of concepts given at training time is incomplete w.r.t. the downstream task. Our method**, as highlighted in the CelebA experiments and the concept subsample CUB experiments in App A.10, **can overcome this difficulty** by learning embeddings that allow information regarding unseen concepts to flow as part of their representation and permit effective concept interventions.
>
> There is a separate interesting question you may be asking about how to handle a setting where not all data points have a complete set of concept annotations. This is a great direction we will explore in future work, thank you!
>
>
> **Clarity of information bottleneck section:** We understand the Reviewer's request for more details as this is certainly an important topic in the literature that has been the subject of several works. While we refer the reviewer to App A.2 (where we provide more details) and the references we indicated in the paper for details, intuitively the way we use such an index in our work is as **a way to measure the information that is preserved at the different levels of the hierarchy of concepts/tasks**. We achieve this by measuring (1) how much information of the input is preserved at the concept-level and (2) how much information of the label is preserved at the concept-level. If a lot of information about the label is lost, then inferring higher-level semantics (i.e., a sample’s label) from concepts could not be possible using concept activations only. Similarly, if a lot of information from the input is lost at the concept level, then we cannot expect that set of concepts to be highly informative of the input, therefore leading to a less interpretable concept space. Hence, we are interested in exploring how these two mutual information quantities vary as training progresses to see if our method is able to circumvent the information bottleneck that has been observed in several works [1].
>
> [1] Shwartz-Ziv, Ravid, and Naftali Tishby. "Opening the black box of deep neural networks via information." arXiv preprint arXiv:1703.00810 (2017).

---

### Official Review · Reviewer_rNZX · 2022-07-19

**Rating:** 7
**Confidence:** 3
**Soundness:** 2 fair
**Presentation:** 3 good
**Contribution:** 3 good

**Summary:**

This paper tackles the trustworthiness of concept bottleneck models (CBM) by improving the accuracy-interpretability tradeoff using a concept-based architecture (Concept Embedding Model or CEM) which represents each concept as a supervised vector. Furthermore, the authors propose two metrics for evaluating concept representations.

**Questions:**

See weaknesses.

**Limitations:**

The authors discussed the limitations of the proposed method in Section 6.

**Strengths And Weaknesses:**

Strengths:
1) The paper tackles an important problem of trustworthiness and accuracy-interpretability tradeoff.
2) The paper is well-organized and easy to follow.
3) The authors performed multiple experiments demonstrating that the proposed approach would work reasonably well in different scenarios and improves the accuracy-interpretability.
4) Scalable to real-world cases with lacking complete concept supervision.

Weaknesses:
1)  Lack of necessary statistical analysis. Some improvements do not seem to be significant. It would be better to provide task and concept accuracy as well as statistical significance, e.g., error bars, in a Table.
2) Does RandInt regularizer increase the training time and cost? I think providing and comparison of the training cost (or model sizes) can also be helpful.
3) Have you tried applying RandInt regularizer to Bool-CBM, Fuzzy-CBM, or other baselines? Based on the provided results in Figure 6, it seems that a lot of improvements are due to adding the RandInt regularizer. For a fair comparison, I would like to see if adding that to the existing methods can improve their performance as well.

---

> ### Author Response · Authors · 2022-08-02
> **Answers to Rev-rNZX**
>
> **Lack of necessary statistical analysis:** We carefully considered the aspects that the Reviewer is mentioning, and opted for a visual representation **to highlight (i) the large, often unforeseen, difference of up to 45% between models’ accuracies and (ii) the task-vs-concept trade-off**. Nevertheless, for completeness we have tabulated these results in Tables 1 and 2 in App A.7 of our updated supplementary material.
>
> **RandInt for other baselines:** This is a really good question. RandInt is a form of regularization that we specifically thought of due to the positive and negative concept embeddings of the proposed model. Its purpose is to incentivize each embedding to be better aligned with the ground truth semantics it represents so that their use in interventions is more effective. When applied to other models, however, it may not have the intended effect. For example, in vanilla CBMs where there is no extra capacity in the bottleneck, RandInt will behave in a similar way to a dropout regularizer and may instead force the label predictor to depend less on a specific concept activation when the concepts are an incomplete description of the task (therefore leading to possibly worse responses to concept interventions). Notice that this does not happen in CEM as during training RandInt still allows gradients to flow and update the weights that generate the “correct” embedding, letting the model modify this embedding so that it is aligned with its intended semantics. On the other hand, if the concepts are a complete description of the downstream task, then, as $p_\text{int}$ approaches 1, we expect RandInt's use in a CBM to behave similarly to how a Independently-trained CBM behaves (where the concept encoder and label predictor models are trained separately). This means that, as shown in [1], it may lead to some improvements in how effective interventions are.
>
> To verify this, we train all CBM baselines with our RandInt regularizer ($p_\text{int} = 0.25$ as in our original work) and include a pointer to these results in Section 5.4 (with all details included in App A.14). We observe in Fig A.11 of App A.14, that, **as hypothesized, RandInt seems in fact to hurt the performance of standard CBMs in concept-incomplete tasks** (e.g., CelebA) while it adds small performance improvements in concept-complete tasks (e.g., CUB). More importantly, however, notice that our main result of our original intervention experiments still hold: CEM still significantly outperforms Hybrid-CBMs, its closest competitor, even when the Hybrid model is trained with RandInt. We thank the reviewer for the insightful question and hope that these experiments answer their concerns.
>
> [1] Koh, Pang Wei, et al. "Concept bottleneck models." International Conference on Machine Learning. PMLR, 2020.

---

> > ### Comment · Reviewer_rNZX · 2022-08-08
> > **Post Rebuttal**
> >
> > The authors' response addressed most of my concerns. I updated my rating to "accept".

---

> > > ### Author Response · Authors · 2022-08-09
> > > **Thank you for your response**
> > >
> > > Dear Reviewer rNZX,
> > >
> > >
> > > We would like to thank you for taking the time to respond to our rebuttal and for updating your score. We are glad to hear that our answers and update have addressed most of your concerns.

---

### Author Response · Authors · 2022-08-02
**Summary for all Reviewers and Area Chairs**

We thank the reviewers for their thoughtful and insightful feedback. It has certainly improved the quality of our manuscript and we hope we are able to address your concerns in this rebuttal and in our updated submission. We reply to questions shared by two or more reviewers in this comment and reply to specific questions reviewers had in comments under their respective feedback.

## Summary of Changes

In our revised version of the paper we have included a few lines to better motivate why we study incorrect interventions and to better explain our choice of baseline models. Furthermore, we have clarified some of the insightful points raised by reviewers in the appendices and added pointers to each appendix in the main paper together with a brief summary of the main results described in each appendix. **However, the core of our work’s contribution and evaluation remains unchanged.** That being said, our changes can be summarized as follows:

- We better motivate why we studied incorrect interventions in Section 5.4.
- Following Rev-6on2’s comment, we fixed a typo in our CAS description ($p$ was used instead of $\rho$). Thank you for pointing this out.
- We emphasize the takeaway of our qualitative results at the end of Section 5.3.
- We extended App A.1 to better motivate our CAS score.
- We included an explanation for our selection of $\gamma$ in App A.4.
- We extended App A.7 to include accuracy and CAS results’ tables and a hypothesis as to why in CUB the “No Concept” model has a high CAS.
- We added App A.8 discussing the computational costs of CEM.
- We added App A.9 discussing the effects of architecture choice in our results.
- We added App A.10 including further experiments on concept subsampling in CUB.
- We added App A.14 showing experimental results of using RandInt for other baselines.

## Answer to common questions

**@Rev-rNZX,Rev-GkfX – Computational cost of CEM and RandInt:** Elaborating on the computational cost of CEM (and of RandInt) is very important and we thank reviewers for bringing this up. We address these concerns in App A.8 and summarize results as follows: Given that our **RandInt regularizer** can be implemented very efficiently as a simple multiplicative Bernoulli mask, in practice (as shown in Fig A.2 (a) of App A.8) we observe it **does not increase the training runtime** (we include a brief comment on this in Section 5.4). Furthermore, we observe that **CEM does not incur significant training convergence times or computational costs compared to other baselines** (see Fig 2 of App A.8). Notice, however, that as mentioned in our conclusion, a training step in CEM does require more FLOPs than vanilla CBMs (we empirically observe less than 10% time increases in large datasets) as its bottleneck is larger and it also performs a series of linear operations on it. Given the performance improvements of our method, we believe that these small computational costs are justified.

Anyway, considering the plots in App A.8 in the revised paper, as the model and datasets become more complex, our method appears not to require increasing time compared to earlier methods.

---

### Meta-Review · Area_Chair_nzeH · 2022-08-25

**Recommendation:** Accept
**Confidence:** Certain

**Metareview:**

This paper proposes Concept Embedding Models, which learn interpretable high-dimensional concept representations to exploit the tradeoff between accuracy, interpretability, and interventions on concepts. Reviewers vote for accepting this paper. The authors are encouraged to further improve this work based on reviewers’ comments in the camera ready and put the new experiments and discussions during the author-reviewer discussion phrase into the final revision, in particular the following:

- Add statistical significance test of experimental results
- Compare training costs and model sizes
- Better justify the proposed CAS mechanism
- Investigate the robustness of learned concepts
- Address the fairness concerns raised by reviewers in comparison with baselines


**Award:**

No

---

### Decision · Program_Chairs · 2022-09-14

Accept